# Bridging Machine Learning and Logical Reasoning by Abductive Learning*

**Wang-Zhou Dai**[†]  **Qiuling Xu**[†]  **Yang Yu**[†]  **Zhi-Hua Zhou**

National Key Laboratory for Novel Software Technology
Nanjing University, Nanjing 210023, China
`{daiwz, xuql, yuy, zhouzh}@lamda.nju.edu.cn`

## Abstract

Perception and reasoning are two representative abilities of intelligence that are integrated seamlessly during human problem-solving processes. In the area of artificial intelligence (AI), the two abilities are usually realised by machine learning and logic programming, respectively. However, the two categories of techniques were developed separately throughout most of the history of AI. In this paper, we present the *abductive learning* targeted at unifying the two AI paradigms in a mutually beneficial way, where the machine learning model learns to perceive primitive logic facts from data, while logical reasoning can exploit symbolic domain knowledge and correct the wrongly perceived facts for improving the machine learning models. Furthermore, we propose a novel approach to optimise the machine learning model and the logical reasoning model jointly. We demonstrate that by using *abductive learning*, machines can learn to recognise numbers and resolve unknown mathematical operations simultaneously from images of simple hand-written equations. Moreover, the learned models can be generalised to longer equations and adapted to different tasks, which is beyond the capability of state-of-the-art deep learning models.

## 1 Introduction

Human cognition [34] consists of two remarkable capabilities: perception and reasoning, where the former one processes sensory information, and the latter one majorly works symbolically. These two abilities function at the same time and affect each other, and they are often joined subconsciously by humans, which is essential in many real-life learning and problem-solving procedures [34].

Modern artificial intelligence (AI) systems exhibit both these abilities. Machine learning techniques such as deep neural networks have achieved extraordinary performance in solving perception tasks [19]; meanwhile, logic-based AI systems have succeeded in human-level reasoning abilities in proving mathematical theorems [27] and in performing inductive reasoning concerning relations [25].

However, popular machine learning techniques can hardly exploit sophisticated domain knowledge in symbolic forms, and perceived information is hard to include in reasoning systems. Even in recent neural networks with the ability to focus on relations [31], enhanced memories and differentiable knowledge representations [13], full logical reasoning ability is still missing—as an example, consider the difficulties of understanding natural language [17]. On the other hand, Probabilistic Logic Program (PLP) [5] and Statistical Relational Learning (SRL) [12] are aiming at integrating learning and logical reasoning by preserving the symbolic representation. However, they usually require semantic-level input, which involves pre-processing sub-symbolic data into logic facts [30].

---

[†]These authors contributed equally to this work.

*W.-Z. Dai (`w.dai@imperial.ac.uk`) and Q. Xu (`simpleword2014@gmail.com`) are now at Imperial College London and Purdue University, respectively.

To leverage learning and reasoning more naturally, it is crucial to understand how perception and reasoning affect each other in a single system. A possible answer is *abduction* [28], or termed as *retro-production* [33]. It refers to the process of selectively inferring specific facts and hypotheses that give the best explaination to observations based on background knowledge [23, 14], where the "observations" are mostly sensory information, and "knowledge" is usually symbolic and structural.

An example of human abductive problem-solving is the decipher-ment of Mayan hieroglyphs [15], which reflects two remarkable human intelligence capabilities: 1) visually perceiving individual numbers from hieroglyphs and 2) reasoning symbolically based on the background knowledge about mathematics and calendars. Fig. 1 shows a Mayan calendar discovered from the Palenque Temple of the Cross Complex, it starts with the mythical *creation date*, followed by a time period written in *long count*, and finished with a specific date encoded by *Tzolk'in* and *Haab'* calendars. Fig. 2 depicts the records of breaking Fig. 1 by Charles P. Bowditch [2]. He first iden-tified some known numbers, and confirmed that the first and sixth hieroglyphs are the same. Then, Bowditch tried substituting those unknown hieroglyphs with visually similar numbers, as shown in "Column 1" in Fig. 2. Meanwhile, he calculated the *Tzolk'in* and *Haab'* values according to his conjectures and background knowl-edge in Mayan calendars, as shown in "Column 2" in Fig. 2. Finally, he got the correct answer "1.18.5.4.0, 1 Ahau 13 Mak" by observing the *consistency* between his conjecture and calculation [2].

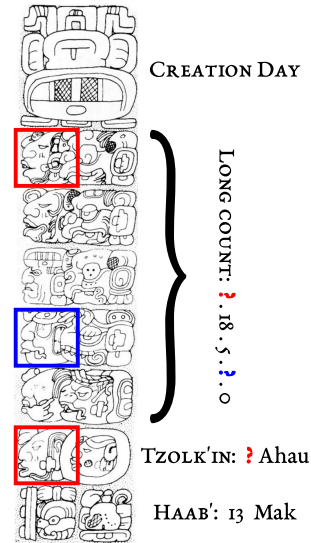

Figure 1: A Mayan calendar. The coloured boxes and "?" cor-respond to unknown numbers.

Inspired by abductive problem-solving, we present the *Abductive Learning* (ABL), a new approach towards bridging machine learning and logical reasoning. In *abductive learning*, a machine learning model is responsible for interpreting sub-symbolic data into primitive logical facts, and a logical model can reason about the interpreted facts based on some first-order logical background knowledge to obtain the final output. The primary difficulty lies in the fact that the sub-symbolic and symbolic models can hardly be trained together. More concretely: 1) it does not have any ground-truth of the primitive logic facts — e.g., the correct numbers in Fig. 1 — for training the machine learning model; 2) without accurate primitive logic facts, the reasoning model can hardly deduce the correct output or learn the right logical theory.

Figure 2: Bowditch's decipherment of Fig. 1 (he wrote "Mak" as "Mac") [2]. Numbers in the vertical boxes are his guesses (Column 1) to the unknown hi-eroglyphs in Fig. 1. The dashed yellow box marks the consistent result accord-ing to his calculation (Column 2).

Our presented *Abductive Learning* (ABL) tries to address these challenges with logical abduction [18, 7] and consistency opti-misation. Given a training sample associated with a final output, logical abduction can conjecture about the missing information — e.g., candidate primitive facts in the example, or logic clauses that can complete the background knowledge — to establish a consistent proof from the sample to its final output. The abduced primitive facts and logic clauses are then used for training the machine learning model and stored as symbolic knowledge, respectively. Consistency optimisation is used for maximising the consistency between the conjectures and the background knowledge. To solve this highly complex problem, we transform it into a task that searches for a function guessing about possibly mistaken primitive facts.

Because of the difficulty of collecting Mayan hieroglyph data, we designed a similar task — the handwritten equation deci-pherment puzzles — for experiments. The task is to learn image recognition (perception) and mathematical operations for calcu-lating the equations (reasoning) simultaneously. Experimental results show that ABL generalise better than state-of-the-art deep learning models and can leverage learning and reasoning in a mutually beneficial way. Further experiments on a visual $n$-queens task shows that the ABL framework is flexible and can improve the performance of machine learning by taking advantage of classical symbolic AI systems such as Constraint Logic Programming [16].

## 2 Related Work

As one of the holy grail problems in AI, combining machine learning and logical reasoning has drawn much attention. Most existing methods try to combine the two different systems by making one side to subsume the other. For example, Fuzzy logic [41], Probabilistic Logic Programming [5] and Statistical Relational Learning [12] have been presented to empower traditional logic-based methods to handle uncertainty; however, most of them still require human-defined symbols as input [30]. Probabilistic programming [35, 21, 20] is presented as an analogy to human cognition to enable probabilistic reasoning with sub-symbolic primitives, yet the correspondence between the sub-symbolic primitives and their symbolic representations used in programming is assumed to already exist rather than assuming that it should be learned.

Another typical approach is to use deep neural networks or other differentiable functional calculations to approximate symbolic calculi. Some of them try to translate logical programs into neural networks, e.g. KBANN [38] and Artur Garcez's works on neural-symbolic learning [10, 9]; others directly replace symbolic computing with differentiable functions, e.g., differential programming methods such as DNC and so on attempt to emulate symbolic computing using differentiable functional calculations [13, 11, 1, 6]. However, few of them can make full-featured logical inferences, and they usually require large amounts of training data.

Different from the previous works, ABL tries to bridge machine learning and logical reasoning in a mutually beneficial way [42]. The two components perceive sub-symbolic information and make symbolic reasoning separately but interactively. The logical abduction with consistency optimisation enables ABL to improve the machine learning model and learn logical theory in a single framework.

## 3 Abductive Learning

In this section, we present the ABL approach. Notations and the problem formulation are firstly introduced, followed by the detailed description and the presented optimisation approach.

### 3.1 Problem Setting

The task of *abductive learning* can be formalised as follows. The input of *abductive learning* consists of a set of labelled training data $D = \{\langle \boldsymbol{x}_1, y_1 \rangle, \ldots, \langle \boldsymbol{x}_n, y_n \rangle\}$ about a target concept $C$ and a domain knowledge base $B$, where $\boldsymbol{x}_i \in \mathcal{X}$ is the input data, $y_i \in \{0, 1\}$ is the label for $\boldsymbol{x}_i$ of target concept $C$, and $B$ is a set of first-order logical clauses. The target concept $C$ is defined with unknown relationships amongst a set of primitive concepts symbols $\mathcal{P} = \{p_1, \ldots, p_r\}$ in the domain, where each $p_k$ is a defined symbol in $B$. The target of abductive learning is to output a hypothesis model $H = p \cup \Delta_C$, in which:

- $p : \mathcal{X} \mapsto \mathcal{P}$ is a mapping from the feature space to primitive symbols, i.e., it is a *perception model* formulated as a conventional machine learning model;

- $\Delta_C$ is a set of first-order logical clauses that define the target concept $C$ with $B$, which is called *knowledge model*.

The hypothesis model should satisfy:

$$\forall \langle \boldsymbol{x}, y \rangle \in D \, (B \cup \Delta_c \cup p(\boldsymbol{x}) \models y). \tag{1}$$

Where "$\models$" stands for logical entailment.

As we can observe from Eq. 1, the major challenge for abductive learning is that the perception model $p$ and the knowledge model $\Delta_C$ are *mutually dependent*: 1) To learn $\Delta_C$, the perception results $p(\boldsymbol{x})$ — the set of groundings of the primitive concepts in $\boldsymbol{x}$ — is required; 2) To obtain $p$, we need to get the ground truth labels $p(\boldsymbol{x})$ for training, which can only be logically derived from $B \cup \Delta_C$ and $y$. When the machine learning model is under-trained, the perceived primitive symbols $p(\boldsymbol{x})$ is highly possible to be incorrect; therefore, we name them *pseudo-groundings* or *pseudo-labels*. As a consequence, the inference of $\Delta_C$ based on Eq. 1 would be inconsistent; when the knowledge model $\Delta_C$ is inaccurate, the logically derived pseudo-labels $p(\boldsymbol{x})$ might also be wrong, which harms the training of $p$. In either way, they will interrupt the learning process.

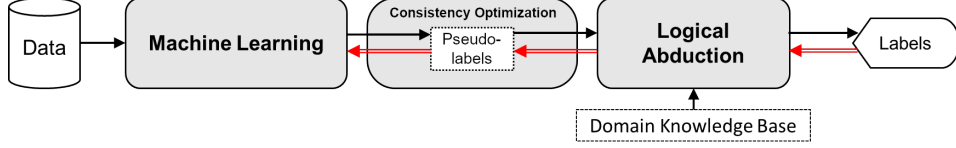

Figure 3: The structure of ABL framework.

## 3.2 Framework

The ABL framework [42] tries to address these challenges by connecting machine learning with an abductive logical reasoning module and bridging them with consistency optimisation. Fig. 3 shows the outline of the framework.

**Machine learning** is used for learning the perception model $p$. Given an input instance $\boldsymbol{x}$, $p$ can predict the pseudo-labels $p(\boldsymbol{x})$ as groundings of possible primitive concepts in $\boldsymbol{x}$. When the pseudo-labels contain mistakes, the perception model needs to be re-trained, where the labels are the revised pseudo-labels $r(\boldsymbol{x})$ returned from logical abduction.

**Logical abduction** is the logical formalisation of abductive reasoning. Given observed facts and background knowledge expressed as first-order logical clauses, logical abduction can abduce ground hypotheses as possible explanations to the observed facts. A declarative framework in Logic Programming that formalises this process is Abductive Logic Programming (ALP) [18]. Formally, an abductive logic program can be defined as follows:

**Definition 1** [18] *An abductive logic program is a triplet* $(B, A, IC)$, *where* $B$ *is* background knowledge, $A$ *is a set of* abducible predicates, *and* $IC$ *is the* integrity constraints. *Given some observed facts* $O$, *the program outputs a set* $\Delta$, *of ground abducibles of $A$, such that:*

- $B \cup \Delta \models O$,
- $B \cup \Delta \models IC$,
- $B \cup \Delta$ *is consistent.*

Intuitively, the abductive explanation $\Delta$ serves as a hypothesis that explains how an observation $O$ could hold according to the background knowledge $B$ and the constraint $IC$.

Considering the formulation in Eq. 1, ABL takes the instance labels about the final concept as observed facts, and takes the hypothesis model $H = \Delta_C \cup p$ as abducibles. Given a fixed $\Delta_C$, ABL can abduce $p(X)$ according to $B$ and $Y$; when the perception model $p$ has been determined, ALP is able to abduce the knowledge model $\Delta_C$ according to $B \cup p(X) \cup Y$. Here we use $p(X) = \bigcup_{i=1}^{n} \{p(\boldsymbol{x}_i)\}$ to represent the pseudo-labels of all the instances $X = \bigcup_{i=1}^{n} \{\boldsymbol{x}_i\}$, and $Y = \bigcup_{i=1}^{n} \{y_i\}$ are the final concept labels corresponding to $X$. Therefore, we can denote the abduced knowledge model conditioned by $B \cup p(X)$ and $Y$ as $\Delta_C(B \cup p(X), Y)$.

## 3.3 Optimisation

The objective of ABL is to learn a hypothesis consistent with background knowledge and training examples. More concretely, ABL tries to maximise the consistency between the abduced hypotheses $H$ with training data $D = \{\langle \boldsymbol{x}_i, y_i \rangle\}_{i=1}^{n}$ given background knowledge $B$:

$$\max_{H = p \cup \Delta_C} \operatorname{Con}(H \cup D; B), \tag{2}$$

where $\operatorname{Con}(H \cup D; B)$ stands for the size of subset $\hat{D}_C \subseteq D$ which is consistent with $H = p \cup \Delta_C$ given $B$. It can be defined as follows:

$$\operatorname{Con}(H \cup D; B) = \max_{D_c \subseteq D} \quad | D_c | \tag{3}$$

$$\text{s.t. } \forall \langle \boldsymbol{x}_i, y_i \rangle \in D_c \ (B \cup \Delta_C \cup p(\boldsymbol{x}_i) \models y_i).$$

To solve Eq. 2, ABL tries to optimise $\Delta_C$ and $p$ alternatively.

During the $t$-th epoch, when the perception model $p^t$ is under-trained, the pseudo-labels $p^t(X)$ could be incorrect and make logical abduction fail to abduce any consistent $\Delta_C$ satisfying Eq. 1, resulting in $\operatorname{Con}(H \cup D; B) = 0$.

Therefore, ABL needs to correct the wrongly perceived pseudo-labels to achieve consistent abductions, such that $\Delta_C^t$ can be consistent with as many as possible examples in $D$. Here we denote the pseudo-labels to be revised as $\delta[p^t(X)] \subseteq p^t(X)$, where $\delta$ is a heuristic function to estimate which pseudo-labels are perceived incorrectly by current machine learning model $p^t$ — in analogy to Bowditch's power of identifying the misinterpreted hieroglyphs (see Fig. 2).

After removing the incorrect pseudo-labels marked by the $\delta$ function, ABL can apply logical abduction to abduce the candidate pseudo-labels to revise $\delta[p^t(X)]$ together with $\Delta_C^t$ by considering:

$$B \cup p^t(X) - \delta[p^t(X)] \cup \Delta_{\delta[p^t(X)]} \cup \Delta_C^t \models Y \qquad (4)$$

Where $p^t(X) - \delta[p^t(X)]$ are the remaining "correct" pseudo-labels determined by $\delta$, and $\Delta_{\delta[p^t(X)]}$ are the abduced pseudo-labels for revising $\delta[p^t(X)]$.

Theoretically, $\delta$ can simply mark all pseudo-labels as "wrong", i.e., letting $\delta[p^t(X)] = p^t(X)$ and ask logical abduction to do all the learning jobs. In this case, ABL can always abduce a consistent $\Delta_{\delta[p^t(X)]} \cup \Delta_C^t$ satisfying Eq. 4. However, this means that the logical abduction have to learn the knowledge model $\Delta_C$ without any influence from the perception model $p$ and the raw data $X$. It not only results in an exponentially larger search space for the abduction, but also breaks the link between logical reasoning and actual data. Consequently, ABL chooses to restrict the revision to be not too far away from the percieved results, by limiting $| \delta[p^t(X)] | \leq M$, where $M$ defines the step-wise search space on the scale of the abduction and is sufficient to be set a small number.

Therefore, when $p^t$ is fixed, we can transform the optimisation problem of $\Delta_C$ into an optimisation problem of function $\delta$, and reformulate Eq. 2 as follows:

$$\max_\delta \quad \text{Con}(H_\delta \cup D), \qquad (5)$$
$$s.t. \quad | \delta[p^t(X)] | \leq M$$

where $H_\delta = p^t(X) - \delta[p^t(X)] \cup \Delta_{\delta[p^t(X)]} \cup \Delta_C^t$ is the abduced hypothesis defined by Eq. 4. Although this objective is still non-convex, optimising $\delta$ instead of $\Delta_C$ allows ABL to revise and improve the hypothesis even when $p^t$ is not optimal.

The heuristic function $\delta$ could take any form as long as it can be easily learned. We present to solve it with derivative-free optimisation [40], which is a flexible framework for optimising non-convex objectives. As to the subset selection problem in Eq. 5, we present to solve it with greedy algorithms.

After obtained the $\delta$ and $\Delta_C^t$, ABL can directly apply logical abduction to obtain the revised pseudo-labels $r(X) = p^t(X) - \delta[p^t(X)] \cup \Delta_{\delta[p^t(X)]}$, which is used for re-training the machine learning model. This procedure can be formulated as follows:

$$p^{t+1} = \arg\min_p \quad \sum_{i=1}^{m} \text{Loss}\left(p(\boldsymbol{x}_i), r(\boldsymbol{x}_i)\right), \qquad (6)$$

where Loss stands for the loss function for machine learning, $r(\boldsymbol{x}_i) \in r(X)$ is the set of revised pseudo-labels for instance $\boldsymbol{x}_i \in X$.

In short, ABL works as follows: Given the training data, an initialised machine learning model is used for obtaining the pseudo-labels, which are then treated as groundings of the primitive concepts for logical reasoning to abduce $\Delta_C$. If the abduction terminated due to inconsistency, the consistency optimisation procedure in Eq. 5 is called to revise the pseudo-labels, which are then used for re-training the machine learning model.

## 4 Implementation

To verify the effectiveness of the presented approach, we designed the handwritten equation decipherment tasks, as shown in Fig. 4 and applied ABL to solve them.

The equations for the decipherment tasks consist of sequential pictures of characters. The equations are constructed from images of symbols ("0", "1", "+" and "="), and they are generated with *unknown* operation rules, each example is associated with a label that indicates whether the equation is correct. A machine is tasked with learning from a training set of labelled equations, and the trained model is

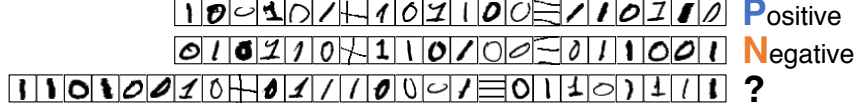

Figure 4: Handwritten equation decipherment puzzle: a computer should learn to recognise the symbols and figure out the unknown operation rules ("xnor" in this example) simultaneously.

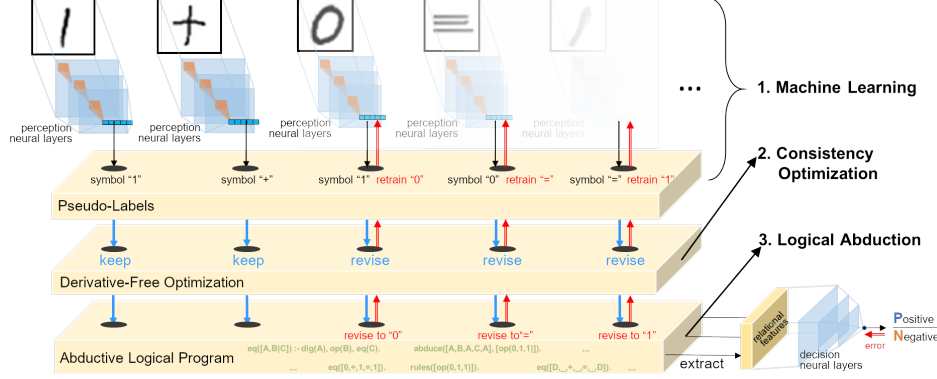

Figure 5: The structure of our ABL implementation.

expected to predict unseen equations correctly. Thus, this task demands the same ability as a human jointly utilising perceptual and reasoning abilities in Fig. 1.

Fig. 5 shows the architecture of our ABL implementation, which employs a convolutional neural network (CNN) [22] as the perception machine learning model. The CNN takes image pixels as input and is expected to output the symbols in the image. The symbol output forms the pseudo-labels. The logical abduction is realised by an Abductive Logic Program implemented with Prolog. The consistency optimisation problem in Eq. 5 is solved by a derivative-free optimisation tool RACOS[40].

Before training, the domain knowledge—written as a logic program—is provided to the ALP as background knowledge $B$. In our implementation, $B$ involves only the *structure* of the equations and a *recursive* definition of *bit-wise operations*. The background knowledge about equation *structures* is a set of definite clause grammar (DCG) rules recursively define that a digit is a sequence of "0" and "1", and each equation share the structure of X+Y=Z, although the length of X, Y and Z may be varied. The knowledge about *bit-wise operations* is a recursive logic program that reversely calculate X+Y, i.e., it operates on X and Y digit-by-digit and from the last digit to the first. The logic programs defining this background knowledge are shown in the supplementary.

**Remark**  Please notice that, the specific rules for calculating the operations are *undefined* in $B$, i.e., results of "0+0", "0+1" and "1+1" could be "0", "1", "00", "01" or even "10". The missing calculation rules form the knowledge model $\Delta_C$, which are required to be *learned* from the data.

After training starts, the CNN will interpret the images to the symbolic equations constructed by pseudo-labels "0", "1", "+" and "=". Because the CNN is untrained, the perceived symbols are typically wrong. In this case, ALP cannot abduce any $\Delta_C$ that is consistent with the training data according to the domain knowledge, i.e., no calculation rules can satisfy the perceived pseudo-labels with the associated labels. To abduce the most consistent $\Delta_C$, ABL learns the heuristic function $\delta$ for marking possible incorrect pseudo-labels.

For example, in the beginning, the under-trained CNN is highly likely to interpret the images as a pseudo-grounding $eq_0$=[1,1,1,1,1], which is inconsistent with any binary operations since it has no operator symbol. Observing that ALP cannot abduce a consistent hypothesis, RACOS will learn a $\delta$ that substituting the "possibly incorrect" pseudo-labels in $eq_0$ with blank Prolog variables, e.g., $eq_1$=[1,_,1,_,1]. Then, ALP can abduce a consistent hypothesis involving the operation rule op(1,1,[1]) and a list of revised pseudo-labels $eq_1$'=[1,+,1,=,1], where the latter one is used for re-train the CNN, helping it distinguish images of "+" and "=" from other symbols.

The complexity of the optimisation objective in Eq. 5 is very high, which usually makes it infeasible to evaluate the entire training set $D$ during optimisation. Therefore, ABL performs abduction and optimisation for $T$ times, each time using a subsample $D_t \subseteq D$ for training. The locally consistent reasoning model $\Delta_C^t$ abduced in each iteration are kept as a relational feature.

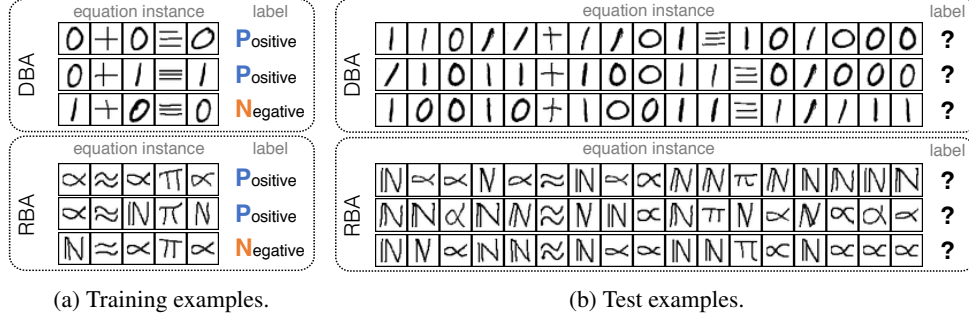

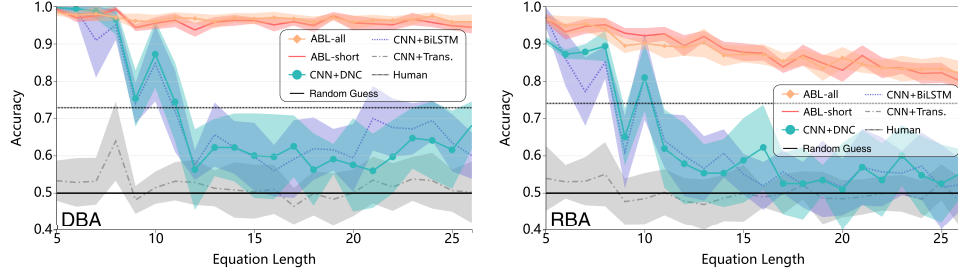

(a) Training examples.  (b) Test examples.

Figure 6: Data examples for the handwritten equations decipherment tasks.

Figure 7: Experimental results of the DBA (left) and RBA (right) tasks.

After the CNN converged or the algorithm meets the iteration limit, all $\langle \boldsymbol{x}_i, y_i \rangle \in D$ are proposition-alised to binary feature vectors by the relational features. For every input equation $\boldsymbol{x}_i$, its pseudo-labels will be evaluated by all the relational features to produce a binary vector $\boldsymbol{u}_i = [u_{i1}, \dots, u_{iT}]$, where

$$u_{ij} = \begin{cases} 1, & B \cup \Delta_C^j \cup p(\boldsymbol{x}_i) \models y_i, \\ 0, & B \cup \Delta_C^j \cup p(\boldsymbol{x}_i) \not\models y_i. \end{cases} \tag{7}$$

Therefore, the original dataset $D = \{\langle \boldsymbol{x}_i, y_i \rangle\}$ can be transforms into a new dataset $D' = \{\langle \boldsymbol{u}_i, y_i \rangle\}$, from which a decision model is learned to handle the noises introduced by subsampling.

## 5 Experiments

**Dataset**  We constructed two image sets of symbols to build the equations shown in Fig. 6. The Digital Binary Additive (DBA) equations were created with images from benchmark handwritten character datasets [22, 36], while the Random Symbol Binary Additive (RBA) equations were constructed from randomly selected characters sets of the Omniglot dataset [21] and shared isomorphic structure with the equations in the DBA tasks. In order to evaluate the perceptual generalisation ability of the compared methods, the images for generating the training and test equations are disjoint. Each equation is input as a sequence of raw images of digits and operators. The training and testing data contains equations with lengths from 5 to 26. For each length it contains 300 randomly generated equations, in a total of 6,600 training examples. This task has $4! = 24$ possible mappings from the CNN outputs to the pseudo-label symbols, and $4^3 = 64$ possible operation rule sets (with commutative law), so the search space of logical abduction contains 1536 different possible $\Delta_C$. Furthermore, the abduction for revising pseudo-labels introduces $2^M$ more candidates. Considering the small amount of training data (especially for the **ABL-short** setting with only 1200 training examples), this task is not trivial.

**Compared methods**

- **ABL**: The machine learning model of ABL consists of a two-layer CNN and a two-layer multiple-layer perceptron (MLP) followed by a softmax layer; the logical abduction will keep 50 calculation rule sets of bit-wise operations set as relational features; The decision model is a two-layer MLP. Two different settings have been tried: the **ABL-all** that uses all training data and the **ABL-short** that only uses training equations of lengths 5-8.

- **Differentiable Neural Computer** (DNC) [13]: This is a deep neural network associated with memory, and has shown its potential on symbolic computing tasks [13].

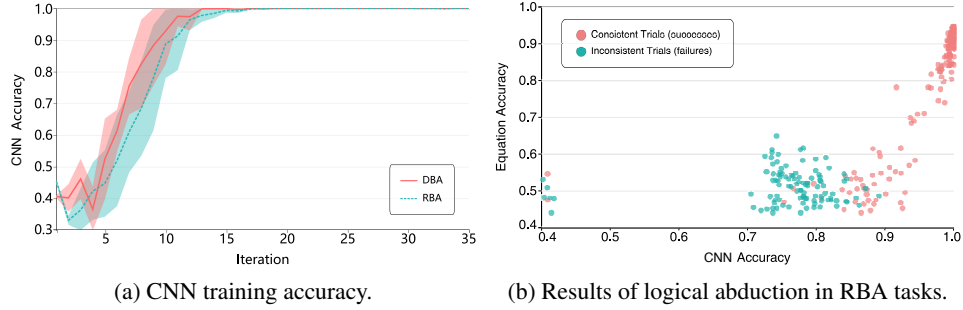

(a) CNN training accuracy.  (b) Results of logical abduction in RBA tasks.

Figure 8: Training accuracy and results of logical abductions.

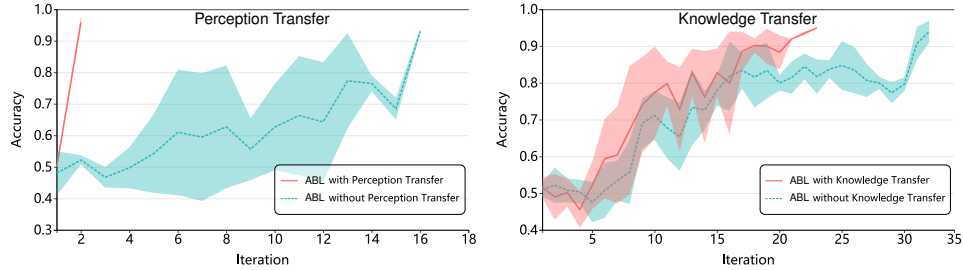

Figure 9: Results of the cross-task transfer experiments.

- **Transformer networks** [39]: This is a deep neural network enhanced with attention, and has been verified to be effective on many natural language processing tasks.

- **Bidirectional Long Short-Term Memory Network** (BiLSTM) [32]: This is the most widely used neural network for learning from sequential data.

To handle image inputs, the BiLSTM, DNC and Transformer networks also use the same structured CNN like the ABLs as their input layers. All the neural networks are tuned with a held-out validation set randomly sampled from the training data. All the experiment are repeated for 10 times and performed on a workstation with a 16 core Intel Xeon CPU @ 2.10GHz, 32 GB memory and a Nvidia Titan Xp GPU.

We also carried out a human experiment. Forty volunteers were asked for classifying images of equations sampled from the same datasets. Before taking the quiz, the domain knowledge about the bit-wise operation was provided as hints, but specific calculation rules are not available — just like the setting for ABL. Instead of using the precisely same setting as the machine learning experiments, we gave the human volunteers a simplified version, which only contains 5 positive and 5 negative equations with lengths ranging from 5-14.

**Results** Fig. 7 shows that on both tasks, the ABL-based approaches significantly outperform the compared methods, and ABL correctly learned the symbolic rules defining the unknown operations. All the methods performed better on the DBA tasks than RBA, because the symbol images in the DBA task are more easily distinguished. The performance of ABL-all and ABL-short have no significant difference, and the performance of the compared approaches degenerates quickly toward the random-guess line as the length of the testing equations grows, while the ABL-based approaches extrapolates better to the unseen data. An interesting result is that the human performance on the two tasks are very close, and both of them are worse than that of ABL. According to the volunteers, they do not suffer from distinguishing different symbols, but machines are better in checking the consistency of logical theories — in which people are prone to make mistakes. Therefore, machine learning systems should make use of their advantages in logical reasoning.

Inside the learning process of ABL, although no ground-truth labels exist for the images of digits and operators, the CNN training accuracy did increase during the learning process, as shown by Fig. 8a. On the other hand, Fig. 8b shows the relationship between ABL's overall equation classification accuracy, image perception accuracy and results of logical abductions on the RBA tasks, where red dots indicate successful abductions and the blue dots signify failures. This result shows that the training of CNN and the logic-based learning of unknown operation rules indeed mutually benefited each other during the training process.

**Cross-task Transfer**    We also carried experiments on transferring the learned CNN and knowledge model (i.e., the relational features $\Delta_C^t$ together with the decision MLP) to different tasks.

The first task transfers the CNN learned from the DBA task to logical exclusive-or equations constructed by the same characters. As shown in Fig. 9, although the final performances of ABLs with and without perception transfer are comparable, the convergence of the ABL with perception transfer is much faster. The second task transfers the learned knowledge model from RBA to DBA domains. As depicted in the right side of the same figure, ABL with knowledge transfer converged significantly faster than the compared method. However, comparing the results between knowledge transfer and perception transfer, we can see that machine learning from sub-symbolic data without explicitly providing the labels is considerably more difficult.

## 6    Discussion

As an important cognitive model in psychology, abduction has already attracted some attention in artificial intelligence [14, 10], while most of existing works combining abduction and induction only consider symbolic domains [37, 7, 29]. There are also some works use abduction to enhance machine learning [4, 24], however, they need to adapt logical background knowledge into functional constraints or use particularly designed operators to support *gradient descent* during learning and reasoning, which relax logical inference into a different continuous optimisation problem.

On the other hand, ABL utilises logical abduction and trial-and-error search to bridge machine learning with original first-order logic, *without* using gradient. As the result, ABL inherits the full power of first-order logical reasoning, e.g., it has the potential of abducing new first-order logical theories that are not in the background knowledge [26]. Consequently, many existing symbolic AI techniques can be directly incorporated without any modification.

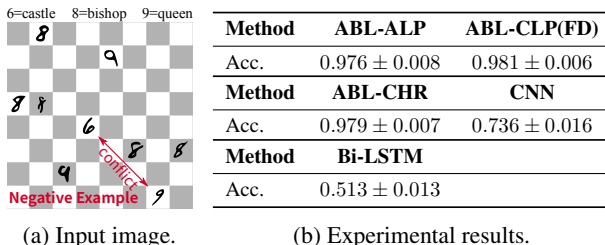

| Method | ABL-ALP | ABL-CLP(FD) |
|---|---|---|
| Acc. | $0.976 \pm 0.008$ | $0.981 \pm 0.006$ |
| **Method** | **ABL-CHR** | **CNN** |
| Acc. | $0.979 \pm 0.007$ | $0.736 \pm 0.016$ |
| **Method** | **Bi-LSTM** | |
| Acc. | $0.513 \pm 0.013$ | |

(a) Input image.          (b) Experimental results.

Figure 10: The extended $n$-queens experiments, $n \in \{2..10\}$.

In order to verify the flexibility of the ABL framework, a further experiment on the *extended $n$-queens task* is shown with Fig. 10, whose inputs are images of randomly generated chessboards consist of blanks, queens, castles and bishops represented by randomly sampled MNIST images, and the associated labels are validity of each board. In this task, we implemented logical abduction with Prolog-based ALP and two popular constraint logic programming [16] systems: Constraint Handling Rules [8] and CLP(FD) [3]. Given recursive first-order logical background knowledge about chess moves, the ABL-based approaches achieved better results comparing to CNN and Bi-LSTM.

## 7    Conclusion

In this paper, we present the *abductive learning*, where machine learning and logical reasoning can be entangled and mutually beneficial. Our initial implementation of the ABL framework shows that it is possible to simultaneously perform *sub-symbolic* machine learning and *full-featured first-order logical reasoning* that allows recursion.

This framework is general and flexible. For example, the perception machine learning model could be a pre-trained model rather than to be learned from scratch; The task for machine learning could be semi-supervised rather than having no label at all; The logical abduction could involve second-order logic clauses to enable abducing recursive clauses and automatically inventing predicates [26]. We hope that the exploration of *abductive learning* will help pave the way to a unified framework accommodating learning and reasoning.

**Acknowledgement**    This research was supported by the National Key R&D Program of China (2018YFB1004300), NSFC (61751306), and the Collaborative Innovation Centre of Novel Software Technology and Industrialisation. The authors would like to thank Yu-Xuan Huang and Le-Wen Cai for their help on experiments, and thank the reviewers for their insightful comments.

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
