[Supplementary Material]

# A   Background Knowledge for Handwritten Equations Decipherment

In this section, we present the background knowledge of the Handwritten Equations Decipherment tasks in Prolog language. Following the convention of logic programming, we use words starting with capital letters and underline dash to represent variables, e.g., "Y", "Pseudo_Label" and "_"; and use numbers and words starting with lowercase to represent constant, predicate an function names, e.g., "0", "+" and "digit/1", where the number behind the slash means the arity of the predicate or function. The sentences initiated with "%" are inline comments.

Table 1: Background knowledge about equation structure.

```
% Define a single digit
digit(0).
digit(1).
% Recursively define digits
digits([D]) --> [D],  digit(D) .
digits([D|T]) --> [D], !, digits(T),  digit(D) .
digits(X) :- phrase(digits(X), X).

% Recursively define equations.
% Since the pseudo-labels may contain missing values (variables),.
% we define eq_args as non operator symbols (including variables).
eq_arg([D]) --> [D],  not(D == '+'), not(D == '=') .
eq_arg([D|T]) -->
    [D], !, eq_arg(T),  not(D == '+'), not(D == '=') .
equation(eq(X, Y, Z)) -->
    eq_arg(X), [+], eq_arg(Y), [=], eq_arg(Z).
parse_eq(Pseudo_Labels, Eq) :-
    phrase(equation(Eq), Pseudo_Labels).
```

Tab. 1 illustrates the background knowledge about digits and equation structures. The predicate `digit/1` defines the two numerical pseudo-labels (primitive concepts) "0" and "1", the other two, "+" and "=", are defined in the Definitive Clause Grammar (DCG) rule of `equation/1`.

The predicates in red colour are *recursive* DCG rules, defining the pattern of corresponding concepts. For example, the rules about `digits/1` indicates that any list longer than 1 and constructed by `digit` are "digits".

The `eq_arg/1` predicate simply defines what are the *argument* of `equation/1`. Because the input pseudo-labels may contain missing values (i.e., the "incorrect" pseudo-labels that have been removed by the learned heuristic $\delta$ function in Eq. 5), the *argument* could be any list composed by `digit` and Prolog's blank variable "_", for example, "10010" and "_101_1".

The DCG rule for `equation/1` defines that any equation has the structure of `eq(X,Y,Z)`, forming a pseudo-label list "`[X],[+],[Y],[=],[Z]`", in which X, Y and Z are instances of `eq_arg/1`.

The predicates in blue simply parses a list given corresponding concepts defined DCG rules, e.g., `parse_eq/2` takes a list of pseudo-labels as input, and outputs the parsed `equation` structure `eq(X,Y,Z)`.

Parsing a sequence with DCG rules is also a kind of *abduction*, in which the DCG rules are *background knowledge*, the input list is *observation*, and the parsed results are *abduced* explanations. Therefore, there could be multiple parsing results. For example, `parse_eq([A,B,C,D,E,F,G],Eq)` will output `Eq=eq([A,B],[D],[F,G])` or `eq([A],[C,D],[F,G])`, where C and E are abduced to be "+" and "=" in the first case; B and E are abduced to be "+" and "=" in the second case.

Tab. 2 shows the background knowledge about bit-wise calculation, which calculates the parsed equation `eq(X,Y,Z)` and abduces the missing pseudo-labels in `eq(X,Y,Z)` as well as the missing operation rules for defining "+". In our implementation, the missing operation rules to be learned, i.e. the $\Delta_C$ is defined with "`my_op/3`". For example, a complete rule set defining arithmetic

Table 2: Background knowledge about bit-wise calculation.

```prolog
% Abductive bit-wise calculation with given pseudo-labels,
% this procedure abduces missing pseudo-labels together with
% unknown operation rules.
calc(Rules, Pseudo) :-
    calc([], Rules, Pseudo).
calc(Rules0, Rules1, Pseudo) :-
    parse_eq(Pseudo, eq(X,Y,Z)),
    bitwise_calc(Rules0, Rules1, X, Y, Z).

% Bit-wise calculation that handles carrying
bitwise_calc(Rules, Rules1, X, Y, Z) :-
    reverse(X, X1), reverse(Y, Y1), reverse(Z, Z1),
    bitwise_calc_r(Rules, Rules1, X1, Y1, Z1),
    maplist(digits, [X,Y,Z]).
% Recursively calculate back-to-front
bitwise_calc_r(Rs, Rs, [], Y, Y).
bitwise_calc_r(Rs, Rs, X, [], X).
bitwise_calc_r(Rules, Rules1, [D1|X], [D2|Y], [D3|Z]) :-
    % Abduces Δ_C (my_op/3) during the calculation.
    abduce_op_rule(my_op([D1],[D2],Sum), Rules, Rules2),
    % Handling carry
    ((Sum = [D3], Carry = []); (Sum = [C,D3], Carry = [C])),
    bitwise_calc_r(Rules2, Rules3, X, Carry, X_carried),
    bitwise_calc_r(Rules3, Rules1, X_carried, Y, Z).
```

binary addition that is going to be learned should contain `my_op(0,0,[0])`, `my_op(0,1,[1])`, `my_op(1,0,[1])` and `my_op(1,1,[1,0])`. However, in the experiments the ABL sometimes output `my_op(1,1,[1])`, `my_op(0,1,[0])`, `my_op(1,0,[0])` and `my_op(0,0,[0,1])`, which flips the semantics of 0 and 1.

The `calc/2` takes a list of pseudo-labels as input, and outputs the possible $\Delta_C$ `Rules` and the missing pseudo-labels that have been removed by the $\delta$ function. `calc/3` function is a more flexible version of `calc/2`, which is able to take some already abduced operation rules `Rules0` in to consideration.

The `bitwise_calc/5` defines the abductive bit-wise calculation process. Given an initialised operation rule set `Rules0` (usually the empty ruleset "`[]`" according to `calc/2`), it abduces the consistent operation rule set `Rules1` with revised pseudo-labels X, Y and Z. Firstly, it calls the `reverse/2` predicate to reverse the equation arguments, then calls the reverse bit-wise calculation predicate `bitwise_calc_r/5` to complete the abduction, and finally check if the abduced pseudo-label forms legitimate `digits`.

The `bitwise_calc_r/5` predicate calculates X+Y to Z bit by bit. It terminates when X or Y runs out of digits and fails if there exists no consistent operation rules with pseudo-labels. During the calculation, it calls the predicate `abduce_op_rule/3` from the Abductive Logic Program (shown in Tab. 3) to abduce consistent operation rules. It also allows 1-digit carry in its calculation.

Tab. 3 defines the Abductive Logic Program (ALP) for the logical abduction in our handwritten equation decipherment tasks.

`abduce/2` is ABL's main predicate for making abduction. Given a set of examples that has been interpreted by the perception machine learning model (and with the learned $\delta$ function marking out the incorrect pseudo-labels), `abduce/2` will try to abduce a $\Delta_C$ and the revised pseudo-labels consistent with all the background knowledge and the labels about the target concept of the examples.

The `abduce/3` predicate processes examples sequentially. By abductively calculating the examples one-by-one, it not along abduces the missing pseudo-labels in each example, but also continuously put consistent bit-wise operation rule in $\Delta_C$ by calling the `calc/3` predicate defined in Tab. 2.

Table 3: The Abductive Logic Program for handwritten equation decipherment.

```
% Main predicate for peforming abduction
% ''Examples'' are the pseudo-labels of a set of examples,
% ''Delta_C'' is the abduced ∆_C.
abduce(Examples, Delta_C) :-
    abduce(Examples, [], Delta_C).
abduce([], Delta_C, Delta_C).
abduce([E|Examples], Delta_C0, Delta_C1) :-
    calc(Delta_C0, Delta_C2, E),
    abduce(Exs, Delta_C2, Delta_C1).

% Getting an existed (already abduced) operation rule from history.
abduce_op_rule(R, Rules, Rules) :-
    member(R, Rules).
% Abduce a new rule.
abduce_op_rule(R, Rules, [R|Rules]) :-
    op_rule(R),
    % integrity constraints.
    valid_rules(Rules, R).

% Integrity Constraints on operation rule set, forbidding
% redundant rules and inconsistent rules.
valid_rules([], _).
valid_rules([my_op([X1],[Y1],_)|Rs], my_op([X],[Y],Z)) :-
    not([X,Y] = [X1,Y1]),
    not([X,Y] = [Y1,X1]),
    valid_rules(Rs, my_op([X],[Y],Z)).
valid_rules([my_op([Y],[X],Z)|Rs], my_op([X],[Y],Z)) :-
    not(X = Y),
    valid_rules(Rs, my_op([X],[Y],Z)).

% Abducing single operation rule.
op_rule(my_op([X],[Y],[Z])) :- digit(X), digit(Y), digit(Z).
op_rule(my_op([X],[Y],[Z1,Z2])) :- digit(X), digit(Y), digits([Z1,Z2]).
```

The `abduce_op_rule/3` called by `bitwise_calc_r/5` can abduce one operation rule in each call. Before doing the abduction, it first returns an already abduced operation rule R in Rules and let `bitwise_calc_r/5` to determine if it is consistent with current calculation. If a history R already meets the requirement then it does nothing; otherwise it will try to abduce a `my_op/3` rule defined by op_rule/1 and return it to the calculation process of `bitwise_calc_r/5`.

During the abduction process, `abduce_op_rule/3` will call an integrity constraint valid_rules/2 to test if the newly abduced R is consistent with previously abduced rule set Rules. Basically, the integrity constraint says that:

1. No redundant operation rules, i.e., there shouldn't be two separate rules defining the same operation X+Y;

2. No conflict operation rules according to the *commutative law*, i.e., X+Y=Y+X.

## B  Errors cases of ABL in the experiments

The failures of ABL-based systems are mostly caused by perception errors. Fig. 11 shows one of the failure examples in the RBA task:

The ground-truth symbols in the equation is "110010+11100110=100011000", but the perceived result by ABL is "110010+11100110110001=000", causing a classification failure. After examined

Figure 11: An example of the wrongly predicted equations during test.

the experimental results, we found that almost all of the learned operation rule sets $\Delta_C^t$ (relational features) are correct, and the equation classification errors are only caused by the incorrectly perceived pseudo-labels. In fact, Fig. 8b has shown that the performance of ABL relies much on perception accuracy. More interestingly, according to the human volunteers, the failures made by them are majorly caused by reasoning errors, i.e. the difficulties in finding consistent operation rules, which is opposite to ABL.