[Reviews · NeurIPS 2019]

Reviewer 1



------------------------------------------------ Comments after reading rebuttal: I've read the rebuttal and appreciate that the authors created a new experiment demonstrating their technique, and which addresses my concern about there only being one demonstration of what is obviously a highly general method. Still, if you can do some version of the Mayan hieroglyphics, or work that example into the introduction, it would improve the paper even more. My score has been raised from 6 to 7. ------------------------------------------------ The paper proposes jointly learning a "perception model" (a neural network), which outputs discrete symbols, along with a "logic model", whose job it is to explain those discrete symbols. They restrict themselves to classification problems, i.e., a mapping from perceptual input to {0,1}; the discrete symbols output by the perception model act as latent variables sitting in between the input and the binary decision. Their approach is to alternate between (1) inferring a logic program consistent with the training examples, conditioned on the output of the perception model, and (2) training the perception model to predict the latent discrete symbols. Because the perception model may be unreliable, particularly early on in training, the logic program is allowed to revise or abduce the outputs of perception. The problem they pose -- integrating learned perception with learned symbolic reasoning -- is eminently important. The approach is novel and intuitive, while still having enough technical ingenuity to be nonobvious. The papers is mostly well-written (although see at the end for a list of recommended revisions). However, they demonstrate their approach on only one very small and artificial feeling problem, when the framework is obviously so generically applicable (for example, could you teach the system to count how many ducks vs geese are in a picture? How about detecting whether a pair of images contain the same object? Or, decoding a sound wave into phonemes and then a word; or playing tic-tac-toe from pixels; or inferring a program from a hand-drawn diagram, or a motor program from a hand-drawn character?). Despite the small-scale nature of the experimental domain, the experiments themselves are excellently chosen, for example they investigate a transfer-learning regime where either the perceptual model for the logical knowledge model is transferred between two variations of their task, and they compare against strong baselines. For these reasons, I weakly recommend that this paper be accepted. Although this paper gets many things right, the one problem is the experiment domain, which feels simple and "toy", and should be ideally supplemented with either another simple domain or a more "real" problem. The supplement does an above-and-beyond job of motivating your domain using Mayan hieroglyphics - definitely put this in the main paper. Can your system solve a version of those Mayan hieroglyphics? Figure 6 suggests that the learned logical theories could be human interpretable. This is great. Can you show some examples of the learned logical rules? Questions for the authors that were unclear in the paper (these should be addressed in a revision): How much of the logical theory can you learn? I browsed through your supplement, and it seems to show that you give a lot, but not all, of the symbolic concepts needed. The paper (lines 200-206) actually seem to imply that even more prior knowledge is given, but the supplement (and the source code to the extent that I browsed it) actually show that what you learned was more impressive (for example, you learned logical definitions of xor, while the text says you give it "a definition of bitwise operations", which I had incorrectly assumed included AND/OR/etc.). What if you were to replace the logical theory learner with something more like metagol - would you get an even more powerful symbolic learner by piggybacking on a system like that? Why do you need the MLP on top of the relational features? Why can't you just use the logical theory and p() at the final iteration? How is equation (3) not equivalent to maximizing the number of examples covered by $B\cup\Delta_C\cup p$ ? Line 148 makes it sound like not satisfying equation 1 means that Con=0, but if I understand it correctly you can both not satisfy equation 1 *and* have Con>0. In equation 5, you have a hard constraint on |\delta|. Why not have a soft constraint, and instead maximize $Con - \lambda |\delta|$, where $\lambda$ would be a hyper parameter? This would get rid of the $M$ hyperparameter. Also, what do you set $M$ to? Can you say something about RACOS? I and many readers will not be familiar with it. Just one qualitative sentence would suffice. Why is RBA so much harder than DBA? Is it only because of the perceptual front-end, i.e. they have the exact same logical structure but different images? Small comments: Figure 7 would benefit from titles on the graphs I wouldn't call your new approach "abductive learning", because "abductive reasoning" is already a widespread term and is easily confused with what you have named your approach. Something like Neurosymbolic Abduction seems both more accurate and has less of a namespace collision with existing terms.

Reviewer 2



* Approaches such as DeepProblog are referenced in the Related Works section, where it is stated that "the structures of first-order logical clauses exploited by these systems are assumed to have already existed" and "they could not obtain new symbolic knowledge during their learning process". I think this point would have deserved more attention, since the image-equation experiment was also conducted for DeepProblog. The difference between the two approaches is not entirely clear, since also the proposed setting relies on a given background logic (though this point is shown only in the supplementary material). Moreover, the experimental section does not properly describes which kind of new symbolic knowledge could be obtained from the proposed approach. * It is clear that fitting the paper within the page limit is not clear, but, in general, many interesting details seem to be in the supplementary material. * The comparison with the results achieve by humans is too briefly explained in the paper, and it is not clear which is the aim of such an experiment. * Some references are missing, about the combination of machine learning and symbolic reasoning in the abductive reasoning scenario. For example, see: -- d'Avila Garcez, Gabbay, Ray, Woods, "Abductive reasoning in neural-symbolic systems", Topoi, 2007 -- Hooldobler, Philipp, Wernhard, "An Abductive Model for Human Reasoning" AAAI Spring Symposium, 2011 Minor corrections: - Pag. 2, "Probabilistic Logic Programming[4]" -> space missing before reference [4] - Pag. 4, "which is consist with" -> "which is consistent with" - Pag. 4, "can consistent with" -> "can be consistent with" - Pag. 7, "For each length it containing" -> "For each length it contains" Originality: moderate, as the novelty with respect to recent neural-symbolic approaches is not entirely clear. Quality: high, sound methodology and experiments. Clarity: high, but could be improved in the experimental section. Significance: moderate, but the research area is very interesting, and this research direction is worth investigating

Reviewer 3



The overall problem of integrating neural and symbolic methods via combining deep learning with deductive, inductive, and abductive logical reasoning is an interesting and important problem, as the authors discuss. The framework that is introduced is interesting and novel and combines deep learning for perception with abductive logical reasoning to provide weakly-labelled training data for the deep-learning perception component. The technique is fairly precisely defined but it was a little hard following all of the notation and equations. It would have been nice to have a sample concrete problem as a example to illustrate the description of the notation and algorithm as it was being discussed in section 3. Waiting until section 4 to see how this abstract formalism was grounded in a concrete problem was a bit frustrating. I find the author's use of the name "abductive learning" for their framework overly broad and insufficiently precise, there has been a range of prior work on using abduction in learning. You should give your work a more descriptive and specific title focusing on the issue of integrating deep learning for perception with abduction and deduction for providing it training data. A lot of other existing work could be called "abductive learning" this term is too general. Particularly, although the paper reviews a number of related works combining machine learning and abduction, there is a range of work from the mid-90's on this topic that is not mentioned, including: Inductive Learning For Abductive Diagnosis, Cynthia A. Thompson and Raymond J. Mooney, In Proceedings of the Twelfth National Conference on Artificial Intelligence (AAAI-94), 664-669, Seattle, WA, August 1994. A collection of relevant papers on the topic from this era is this book: P. A. Flach and A. C. Kakas, editors, Abduction and Induction, 2000. Kluwer Academic This older work could be cited and discussed. Some details of the method I found confusing and/or limiting . When introduced, the abduced knowledge Delta_c is described as a "set of first-order logical clauses" when eventually it seemed to be clear that these could only be ground literals, not general clauses. The system for abductive logic programming can only abduce a set of ground literals as assumptions, not general clauses. Therefore, the only symbolic knowledge it can learn is specific ground literals, not general quantified clauses (e.g. Horn rules). The fixed knowledge, B, which apparently must be prespecified and cannot be learned or modified by learning, must be carefully crafted to allow the abducibles to represent the requisite symbolic knowledge to be learned as a set of ground literals. This seems very limiting and requires carefully hand-crafting the actual symbolic knowledge B which cannot be learned. How could B be automatically learned or revised? The parameter M, seems like a fairly ad-hoc hyper-parameter which must be manually tuned for a particular problem. How is this parameter set? The test problem of visual equation classification seems very contrived and very toy. It seems the knowledge based B had to be carefully crafted just for this specific problem and this knowledge cannot be learned or modified. It would be good to show the actual clauses in B, I would have liked to have seen application of the approach to a more realistic, real-world problem rather than this single, highly-artificial problem. Overall, I am mildly recommending accept since I think integrating neural and symbolic learning and reasoning is a very important problem and, despite its limitations, the paper presents an interesting new approach for doing this.

[Author Response · NeurIPS 2019]

1    We want to thank the reviewers for the supportive and insightful comments, and we will improve the paper accordingly.

2    In the following we focus on the major technical questions because of the page limit.

3    **To Reviewer #1**:

4    **Q1**    The technique is very general ... the one problem is the experiment domain, which feels simple and "toy", and

5    should be ideally supplemented with either another simple domain or a more "real" problem. *(Also asked by Rivewer 3)*

Figure 1: Extended $n$-queens and the results.

| | ABL-ALP | ABL-CLP-FD |
|---|---|---|
| | $0.976 \pm 0.008$ | $0.981 \pm 0.006$ |
| | **ABL-CHR** | **CNN** |
| | $0.979 \pm 0.007$ | $0.7355 \pm 0.016$ |
| | **Bi-LSTM** | |
| | $0.513 \pm 0.013$ | |

**A1**    We agree that the proposed framework is general and applicable to many tasks. Therefore, we have made a quick experiment — the *extended $n$-queens task*, whose inputs are images of chess boards with blanks, queens, castles and bishops (represented by randomly sampled MNIST images), and the labels are whether the state of board is valid. Furthermore, we implemented logical abduction with plain ALP and two popular *constraint logic programming* systems (CHR & CLP-FD). An example and the results are shown in fig. 1.

15    **Q2**    What if you were to replace the logical theory learner with something more like metagol?

16    **A2**    Thank you for the excellent suggestion. Metagol is implemented by second-order abduction, which fits in our

17 proposed framework very well; it will also enable inducing recursive first-order logical rules and predicate invention.

18 There are many other options, such as constraint logic programming, answer set programming, and so on, as long as

19 they can make logical abduction. Meanwhile, this paper focuses on presenting and verifying the idea of combining

20 perception with reasoning by abduction and trial-and-error search. Thus we will discuss the above as the future work.

21    **Q3**    Why do you need the MLP? ... Why can't you just use the logical theory and $p()$ at the final iteration?

22    **A3**    We add MLP because the logical theories abduced from *subsamples* of the data might be inaccurate; and the

23 labels may also contain noise. MLP is convenient for dealing with these noise. We will revise to discuss this point.

24    **Q4**    Why not have a soft constraint (in eq. 5)? What do you set $M$ to? Is it ad-hoc? *(Also asked by Rivewer 3)*

25    **A4**    In the experiments, we set $M$ equals to 10. $M$ defines the step-wise search space on the scale of the abduction

26 and is sufficient to be set a small number. We will revise to discuss this.

27    **Q5**    Can you say something about RACOS?

28    **A5**    It is a derivative-free optimisation approach like evolutionary algorithm with theoretical ground.

29    **Q6**    Why is RBA so much harder than DBA? Is it only because of the perceptual front-end?

30    **A6**    It is because of the perceptual front-end, the datasets share the same equation structures. RBA images are more

31 difficult for the CNN network to distinguish, because it doesn't have a large amount of training data.

32    **To Reviewer #2**:

33    **Q1**    The difference between the proposed approach to DeepProbLog is not entirely clear.

34    **A1**    DeepProbLog relies on Algebraic Prolog that has particularly designed operators to *support gradient descent*

35 for jointly optimising the parameters of ProbLog programs and the neural nets; While ABL utilises logical abduction

36 and trial-and-error search to bridge the neural nets with the original Prolog system, *without* using gradient. As the

37 result, our framework inherits the full power of first-order logical reasoning, e.g., it can abduce new logical theories that

38 are not in the background knowledge. Consequently, many existing symbolic AI techniques, such as constraint logic

39 programming, can be directly incorporated in this framework *without* introducing gradients—usually by relaxing logic

40 inference to continuous functions. Thank you for pointing out the issue, we'll add more discussion in the revision.

41    **Q2**    ... many interesting details seem to be in the supplementary material. *(Also asked by Rivewer 1)*

42    **A2**    Thank you and Reviewer 1 for the nice suggestion, we will re-arrange the paper structure by moving the motivating

43 example and the details about the background knowledge to the main text.

44    **Q3**    The comparison with the results achieved by humans is to briefly explained.

45    **A3**    We will revise to explain the results. Roughly speaking, it shows that human do not suffer from distinguishing

46 different symbols, while machines are better in checking the consistency of logical theories — in which people are

47 prone to make mistakes. Therefore, machine learning systems should make use of their advantage in logical reasoning.

48    **To Reviewer #3**:

49    **Q1**    ... the abduced knowledge $\Delta_C$ is described as "a set of first-order logical clauses" when eventually it seems to be

50 clear that these could only be ground literals. *(Also asked by Rivewer 1)*

51    **A1**    In the equation decipherment tasks, abducing groundings is sufficient for learning the binary operations, so we

52 implemented the abduction with first-order logic. However, our framework does not restrict the order of background

53 knowledge because it utilises the original Prolog system. By exploiting 2nd-order abduction like metagol (as suggested

54 by Reviewer 1), it could learn more complicated theories. We will revise to discuss the above and make it clear.

55    **Q2**    The fixed knowledge $B$ cannot be modified by learning. It would be good to show the actual clauses in $B$.

56    **A2**    The programs in the supplementary material are the actual clauses in our codes. They only describe the general

57 structure of the domain, while the more concrete logical theories are learned. How to modify the general background

58 knowledge, i.e., non-monotonic reasoning, is a significant open problem in symbolic AI. The formulation of our method

59 is flexible, thus it is possible to incorporate techniques like *belief revision* to deal with this problem in the future.

[Meta-Review · NeurIPS 2019]

The reviewer consensus was that, despite requiring some improvements in terms of presentation, with some areas flagged by reviewers as necessitating more detail, and the toy-ish nature of the experiments, that this paper addresses an important problem with the NeurIPS community in attempting to reconcile deep networks with symbolic-like reasoning. The paper is thus deemed of an acceptable standard, but the authors should note that while they are not expected to change their experimental setting for the camera-ready, should the paper be included in the proceedings, they should pay careful attention to the reviewer comments and recommendations when revising their paper in order to insure that the points of clarification requested are expanded upon, possibly in an appendix.